# COVID-19 and influenza infections mediate distinct pulmonary cellular and transcriptomic changes

Chenxiao Wang [1,2,6], Mst Shamima Khatun[3,6], Zhe Zhang [4], Michaela J. Allen[4], Zheng Chen[1,2], Calder R. Ellsworth[1,2], Joshua M. Currey[1,2], Guixiang Dai[3], Di Tian[5], Konrad Bach[5], Xiao-Ming Yin [5], Vicki Traina-Dorge[1,2], Jay Rappaport[1,2], Nicholas J. Maness [1,2], Robert V. Blair [1], Jay K. Kolls [3], Derek A. Pociask [4✉] & Xuebin Qin [1,2✉]

SARS-CoV-2 infection can cause persistent respiratory sequelae. However, the underlying mechanisms remain unclear. Here we report that sub-lethally infected *K18*-human *ACE2* mice show patchy pneumonia associated with histiocytic inflammation and collagen deposition at 21 and 45 days post infection (DPI). Transcriptomic analyses revealed that compared to influenza-infected mice, SARS-CoV-2-infected mice had reduced interferon-gamma/alpha responses at 4 DPI and failed to induce keratin 5 (*Krt5*) at 6 DPI in lung, a marker of nascent pulmonary progenitor cells. Histologically, influenza- but not SARS-CoV-2-infected mice showed extensive Krt5+ "pods" structure co-stained with stem cell markers Trp63/NGFR proliferated in the pulmonary consolidation area at both 7 and 14 DPI, with regression at 21 DPI. These Krt5+ "pods" structures were not observed in the lungs of SARS-CoV-2-infected humans or nonhuman primates. These results suggest that SARS-CoV-2 infection fails to induce nascent Krt5+ cell proliferation in consolidated regions, leading to incomplete repair of the injured lung.

[1] Tulane National Primate Research Center, Covington, LA 70433, USA. [2] Department of Microbiology and Immunology, Tulane University School of Medicine, New Orleans, LA 70112, USA. [3] Department of Medicine and Pediatrics, Center for Translational Research in Infection and Inflammation, Tulane University School of Medicine, New Orleans, LA 70112, USA. [4] Department of Pulmonary Critical Care and Environmental Medicine, Tulane University School of Medicine, New Orleans, LA 70112, USA. [5] Department of Pathology and Laboratory Medicine, Tulane University School of Medicine, New Orleans, LA 70112, USA. [6]These authors contributed equally: Chenxiao Wang, Mst Shamima Khatun. ✉email: dpociask@tulane.edu; xqin2@tulane.edu

Prolonged COVID lung symptoms including fatigue, cough, and shortness of breath can persist for months. The residual lung abnormalities such as fibrosis were estimated in up to 11% of people discharged after COVID-19-related hospitalization[1,2]. Within 8 months of discharge after COVID-19 hospitalization, pulmonary abnormalities were still visually identifiable on thoracic follow-up CT scans[1]. These residual lung abnormalities minimally declined per week after discharge, and minimal resolution was observed in paired subsequent scans at least 90 days apart[1]. One-fourth of long COVID patients had lung zones with ≥10% presence of ground-glass opacities at three months post-discharge, and one out of five follow-up patients presented with parenchymal bands associated with early progression of fibrosis[3]. These changes are not limited to severe patients as there is evidence showing mild to moderate COVID patients also developed long respiratory COVID sequelae after clearing SARS-CoV-2 (CoV2) shedding[4]. Fatigue, respiratory syndrome, and chest X-ray abnormalities were found in both intensive care unit (ICU) and non-ICU hospitalized patients at least 6 weeks post-acute phase[5].

The recent emergence of highly pathogenic coronaviruses has resulted in only a limited number of studies regarding the mechanisms involved in COVID-associated scarring and fibrosis. However, insight may be gained by comparing COVID infection to highly pathogenic H1N1 Flu infection. Flu infection causes epithelial changes as far out as 60 days but with limited scarring[6]. This ability to recover lung capacity after Flu infection has been demonstrated in both in vivo and in vitro studies[7–10]. This recovery involves multipotent basal cells that express keratin 5[11–13]. During infection, basal cells proliferate and form nascent, discrete Krt5+ "pods" in the consolidation lung regions after Flu infection and contribute to epithelial recovery in the lung parenchyma[12,14]. However, it is unclear whether Krt5+ cells are induced and expanded after acute CoV2 infection.

To understand the mechanisms of lung recovery after acute CoV2 infection, we characterized a chronic COVID mouse model using a sub-lethal dose of CoV2 to infect K18 mice[15,16] and compared their recovery with Flu-infected mice. We found that CoV2 and Flu infections mediate distinct pulmonary cellular and transcriptomic changes. CoV2, but not Flu infection fails to induce Krt5+ progenitor cell proliferation, leading to more persistent and chronic pulmonary consolidation and scarring than other respiratory viral infections.

## Results

**K18 mice infected with a sub-lethal dose of CoV2 have persistent lung abnormalities.** The K18 transgenic mice express human ACE2, a CoV2 receptor under the transcriptional control of the human cytokeratin 18 promoter in airway epithelial cells[17–21]. We have shown that the K18 mice infected with a lethal dose ($2 \times 10^5$ TCID50) of CoV2 develop severe acute pneumonia, and that all infected mice meet euthanasia criteria within 7 days post-infection (DPI)[15]. To examine the chronic impact of CoV2 infection on the lung, we infected the mice with a twenty-fold lower ($1 \times 10^4$ TCID50) inoculum of CoV2 (Fig. 1a) (detailed mice information are shown in Supplementary Table 1). Both lethal and sub-lethal infections induced loss of body weight starting at 4 DPI (Fig. 1a); however, K18 mice infected with the sub-lethal dose of CoV2 failed to reach euthanasia criteria, regaining body weight starting at 8 DPI, and reaching the original body weight by 13 DPI (Fig.1a). We detected viral genomic RNA in the lungs of sub-lethal dose infected K18 mice at 21 DPI (Fig.1a). To further monitor the replication of CoV2, we assayed sub-genomic viral RNA load[15,22]. There were no detectable CoV2 sub-genomic viral RNAs in the lungs or brains of the infected K18 mice at 21 DPI (Fig. 1c). Hematoxylin and eosin (H&E) staining showed patchy consolidation in all three K18 mice at 21 DPI (Fig.1d) and two K18 mice at 45 DPI infected with sub-lethal dose (Supplementary Fig. 1a–h). In the nasopharyngeal cavity of infected K18 mice at 3 DPI, we found that SARS-CoV-2 S protein co-localized with lung epithelial marker Pan cytokeratin (PanCK)[23] (Supplementary Fig. 2a). This result further demonstrates that SARS-CoV-2 initially infects the epithelial cells of the large airways in this infected K18 mice model. Immunofluorescent (IF) staining showed the co-localization of CoV2 spike protein with surface macrophages marker CD206 on lungs of sub-lethal CoV2 infected mice at 21 DPI (Fig. 1e). The co-localized cells take up about 80% among CD206 macrophages in these three sub-lethally infected mice (Fig. 1f).

**Lung transcriptomic analysis of acute and chronic CoV2 infection in K18 mice.** To determine the transcriptional changes from acute, severe disease to chronic lung injury, we collected RNA from the lungs of CoV2-infected K18 mice at 21 DPI and performed bulk RNAseq analysis. K18 mice inoculated with phosphate-buffered saline were used as naïve control. Results were also compared with RNAseq analysis from K18 mice infected with a lethal CoV2 inoculum at 4 DPI from our previous study[15]. Compared to the naïve control, CoV2-infected K18 mice had increased expression of several collagen superfamily genes including *Col1a1, Col28a1, Col5a2, Col3a1, Col15a1, Col1a2, Col5a1, Col6a3, Col4a5, Col27a1, Col8a1, Col7a1, Col16a1, Col4a4*, and other extracellular matrix protein genes such as elastin gene *Eln* and clusterin gene *Clu* in lungs at 21 DPI (Fig. 2a, Supplementary Data 1, 2). Compared to infected K18 mice at 4 DPI, CoV2-infected K18 mice also had increased *Col15a1* expression in the lung at 21 DPI (Fig. 2b, Supplementary Data 3). Moreover, we found evidence for higher activation of epithelial-to-mesenchymal transition pathways (Fig. 2c,d, and Supplemental Fig. 2b) at 21 DPI compared to lung tissue from naïve or infected K18 mice at 4 DPI. Furthermore, compared to lungs of naïve K18 mice, CoV2-infected K18 mouse lungs at 21 DPI also had increased activation of hallmark inflammatory response and interferon-gamma pathways in addition to increased complement pathways and Kras signaling (Fig. 2c). Since the production of collagens, elastin and clusterin play an essential role in fibrotic responses after tissue injury[24–26], these results indicate that sub-lethal infection with CoV2 induces a chronic inflammatory response and may mediate fibrosis in the lungs of K18 mice at 21 DPI.

**Lung transcriptomic analysis of CoV2 and A/PR/8/34 (H1N1)-infected K18 mice.** Chronic lung injury from viral infection is not unique to CoV2. In a previous publication, pulmonary lesions were shown to persist for 60 days during experimental influenza A/PR/8/34 (H1N1) infection[6]. However, unlike the fibrotic state induced by COVID, flu-induced transcriptomic and epigenetic changes are more often associated with repair[6]. To identify unique gene signatures that may drive the lung from a reparative state to a fibrotic state, we conducted bulk RNA sequencing analysis from the lungs of K18 mice infected with either CoV2 ($2 \times 10^5$ TCID50) or H1N1 influenza (A/PR/8/34, 50 PFU)[6,15,27] at 4 or 6 DPI and compared them with these obtained from Naïve K18 mice. A heatmap of these findings shows that at 4 DPI and 6 DPI compared with Naïve K18 mice, both Flu and COV2 infections significantly induced a greater number of pro-inflammatory genes such as *Saa3* and chemokines such as *Cxcl11, Cxcl10, Ccl, Ccl2, Ccl7*. We also observed that Flu infection mediated higher induction of those gene expressions than CoV2 infection (Fig. 3a,b, Supplementary Data 4, 5).

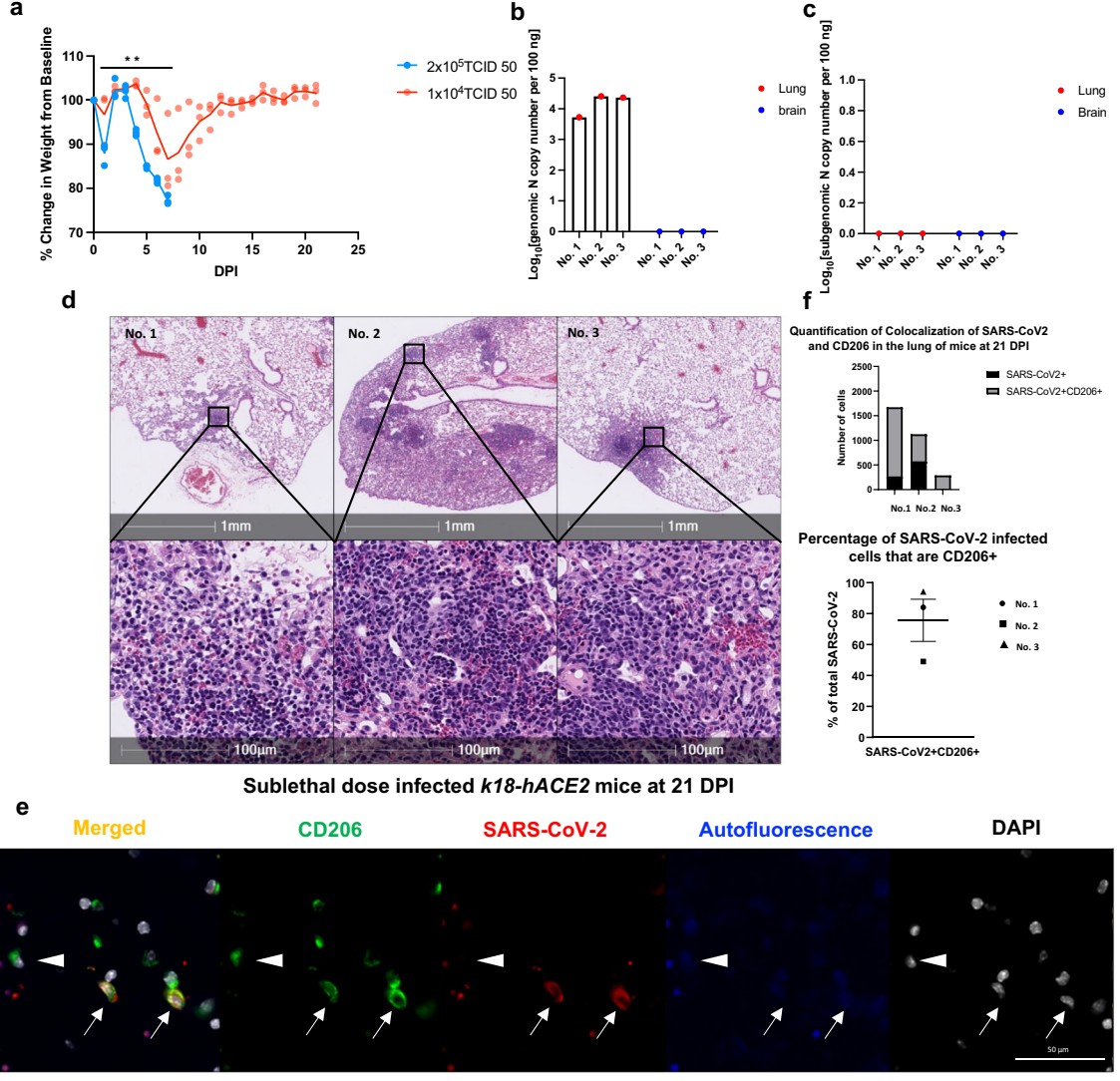

**Fig. 1 K18 mice infected with a sub-lethal dose of SARS-CoV-2 sustain chronic lung injury (n = 3). a** Male and female *K18* mice were infected with $1 \times 10^4$ TCID$_{50}$ (n = 3) or $2 \times 10^5$ TCID$_{50}$ (n = 3) of SARS-CoV-2 intranasally and body weight changes were monitored. P < 0.01 vs $2 \times 10^5$ TCID$_{50}$ by two-way ANOVA. Data are presented as mean and individual values. Total viral genomic N mRNA (**b**) and subgenomic N mRNA (sgmRNA) levels (**c**) in the lungs and brains of the *K18* mice infected with a sub-lethal dose of SARS-CoV-2 ($1 \times 10^4$ TCID$_{50}$) and euthanized at 21 days post infection. The level of sgmRNA in the various organs were measured by qRT-PCR. **d** H&E images of lung sections show persistent, patchy pneumonia in the sub-lethal dose-infected *K18* mice at 21 DPI. **e, f** Representative immunofluorescence staining of CD206 (green) and SARS-CoV-2 spike protein(red) in lungs from sub-lethal SARS-CoV-2 dose-infected *K18* mice at 21 DPI. Nuclei are stained with DAPI (white). Staining demonstrates CD206 expression with (arrows) and without (arrowhead) SARS-CoV-2 spike protein. **f** Quantitative analysis of co-localization of CD206 macrophages and spike protein in the lungs shows the majority of SARS-CoV-2 positive cells in all three animals are CD206+ macrophages based on absolute number and percentage of total SARS-CoV-2 positive cells.

Differential gene expression was also analyzed from Flu and CoV2-infected *K18* mice at 4 or 6 DPI (Fig. 3c,d, Supplementary Data 6, 7). At 4 DPI, both infections induced similar weight loss. However, compared to CoV2 infection, Flu infection significantly induced a greater number of pro-inflammatory genes (*Irg1, Saa3, Il6*) and chemokines (*Cxcl11, Cxcl10, Ccl7, Ccl2, Cxcl2, Ccl1*) (Fig. 3c, Supplementary Data 6). Pathway analysis revealed a greater induction of type I and type II interferon (IFN) pathways, as well as TNF alpha signaling via NFκB pathways in Flu infected mice compared to CoV2-infected mice at 4 DPI (Fig. 3e and Supplemental Table 2). TNF alpha signaling via the NFκB pathway and the inflammatory response pathways remained significantly elevated through day 6. In addition, Flu infection also significantly induced higher activation of P53, Kras signaling, epithelial-mesenchymal transition, and apical junction pathways

compared to CoV2 infection at 6 DPI (Fig. 3f and Supplemental Table 3). Moreover, three keratin genes (*Krt16, Krt15, Krt5*) were also significantly induced in Flu-infected mice, suggesting epithelial changes (Fig. 3d). Of particular interest was *Krt5*, which is a marker for lung progenitor basal cell keratin 5, a protein that has been associated with Flu injury and repair[14,28].

**Krt5+ cells are not found in pulmonary consolidation regions during CoV2 infection.** The finding that CoV2 infection had substantially lower expression of *Krt5* transcripts at 6 DPI prompted us to investigate whether this was due to a failure of Krt5+ cells to expand in CoV2 infection compared to influenza infection[12–14]. We performed IF staining to compare the number of Krt5+ cells at 7, 14, and 21 DPI in CoV2 and Flu infected lungs (Fig. 4a and Supplementary Fig. 3). To monitor interstitial changes

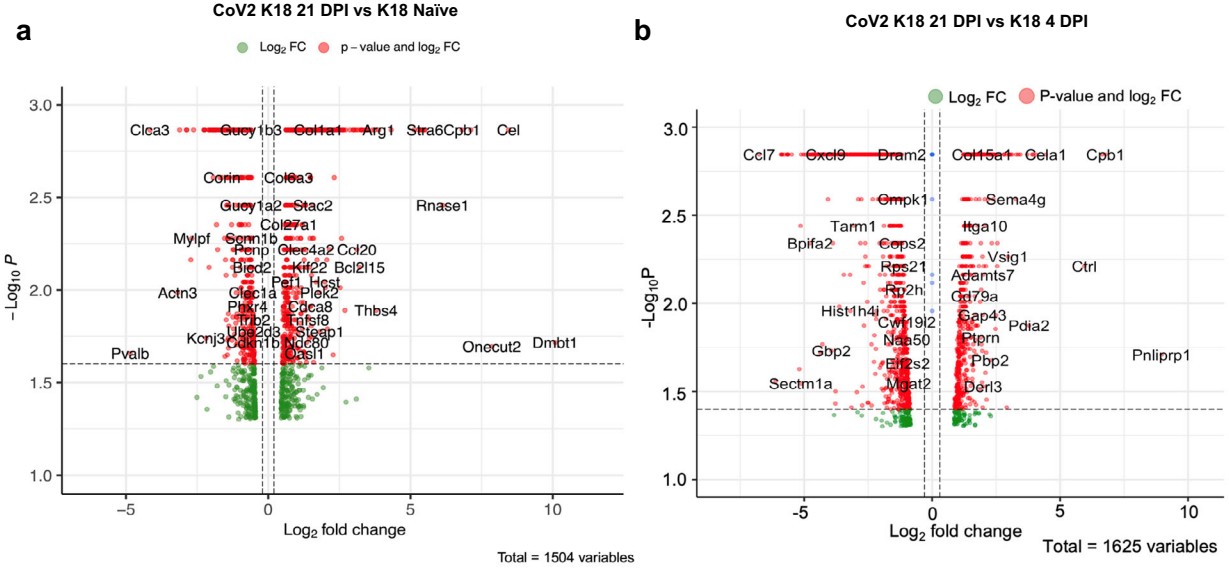

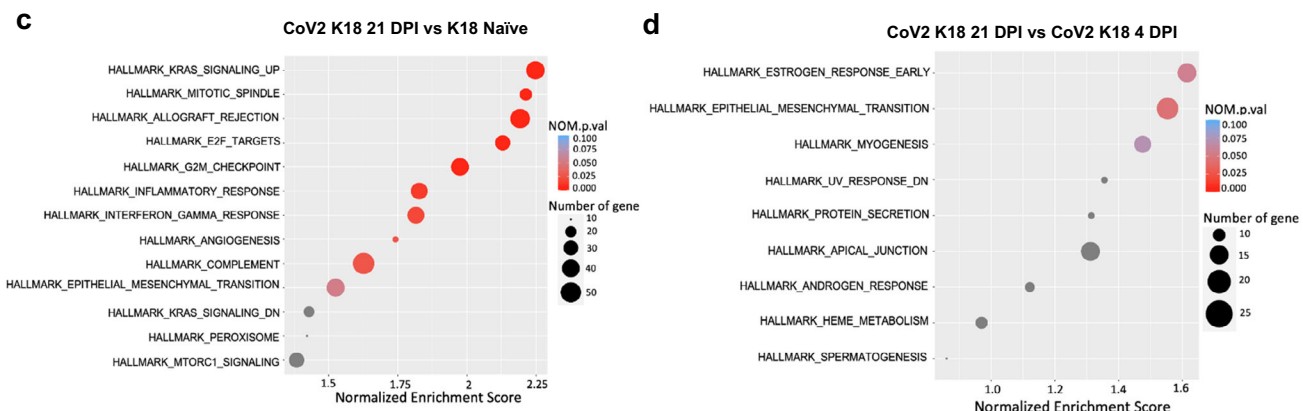

**Fig. 2 Transcriptomic analysis of the lungs of *K18* mice infected with SARS-CoV-2 at 21 DPI and without infection, and of *K18* mice infected with SARS-CoV-2 at 4 DPI.** Enhanced volcano plot of differentially expressed genes (DEGs) (**a**) and pathways (**c**) between sublethal SARS-CoV-2 dose-infected *hACE2-K18* ($1 \times 10^4$ TCID$_{50}$) lungs (CoV K18 21 DPI) at 21DPI ($n = 3$) and *K18* Naïve mice. Enhanced volcano plot of DEGs (**b**) and pathways (**d**) between sublethal dose SARS-CoV-2 infected *K18* lungs at 21DPI (CoV K18 21 DPI) and lethal dose SARS-CoV-2-infected *K18* lungs ($2 \times 10^5$ TCID$_{50}$) at 4DPI (CoV K18 4 DPI).

and fibrosis in consolidated lung regions, we co-stained for alpha smooth muscle actin (SMA) (Fig. 4a,b). Scattered Krt5 positive cells were found in Flu infected mice through 7–21 DPI, peaking at 14 DPI as defined by the highest percentage of proliferative Krt5+ cells and largest number of discrete Krt5+ pods (Fig. 4a–d and Supplementary Fig. 3). In striking contrast, we did not observe any Krt5+ "pods" or Krt5+ cells in consolidated areas in COVID lungs (Fig. 4a,c,d). Consolidated areas in Flu and COVID mouse lungs were shown to exhibit a mild increase in SMA expression by 14 DPI that resolves in Flu infected mice but persists at a low level by 21 DPI in CoV2-infected mice (Fig. 4a,b). We also found that Krt5+ cells express both transformation-related protein 63 (Trp63) and nerve growth factor receptor (NGFR), both pulmonary stem cell markers[12,14,29,30] (Supplementary Figs. 4a,b, 5), which further confirms the airway stem cells. Moreover, we documented that Krt5+ "pod" structures contain dense AT1 cells identified by E-Cad staining, and scattered AT2 cells identified by pro-surfactant protein C (pro-SPC) (Supplementary Fig. 6). Together, these data show that Krt5+ progenitor cells proliferate in Flu-infected lung but are absent of injury in CoV2 infected lung.

We further used Picrosirius red (PSR) staining to investigate pulmonary collagen deposition between CoV2 and Flu infected lungs at different time points post infection (Fig. 5, Supplementary Fig. 7). We observed that CoV2-infected mice tended to increase in collagen deposition in the lungs as compared to Flu infected mice at 21 DPI only, but not other time intervals (Fig. 5a,b). In contrast, the flu-infected lungs had minor collagen deposition across the disease time course. The dynamics of influenza-induced collagen deposition correlate with the disease progression and recovery. Viral load peaks usually around day 5 which is followed by severe inflammation and disease (weight loss) from days 7–10. Injury peaks from 10–18 DPI and resolves 20–30 DPI (Fig. 5a,b)[6]. Collagen deposition in CoV2-infected lungs was sustained at 7, 14, 21, and 45 DPI (Fig. 5a) and 45 DPI. Similar findings were observed with trichrome staining which showed that CoV2-infected mice had significantly higher collagen deposition in the lungs than Flu infected mice (Supplementary Fig. 1e,h and Supplementary Figs. 8, 9). Together, these results demonstrate that there is a sustained collagen deposition in CoV2 but not Flu infected lungs and support the notion that sub-lethal

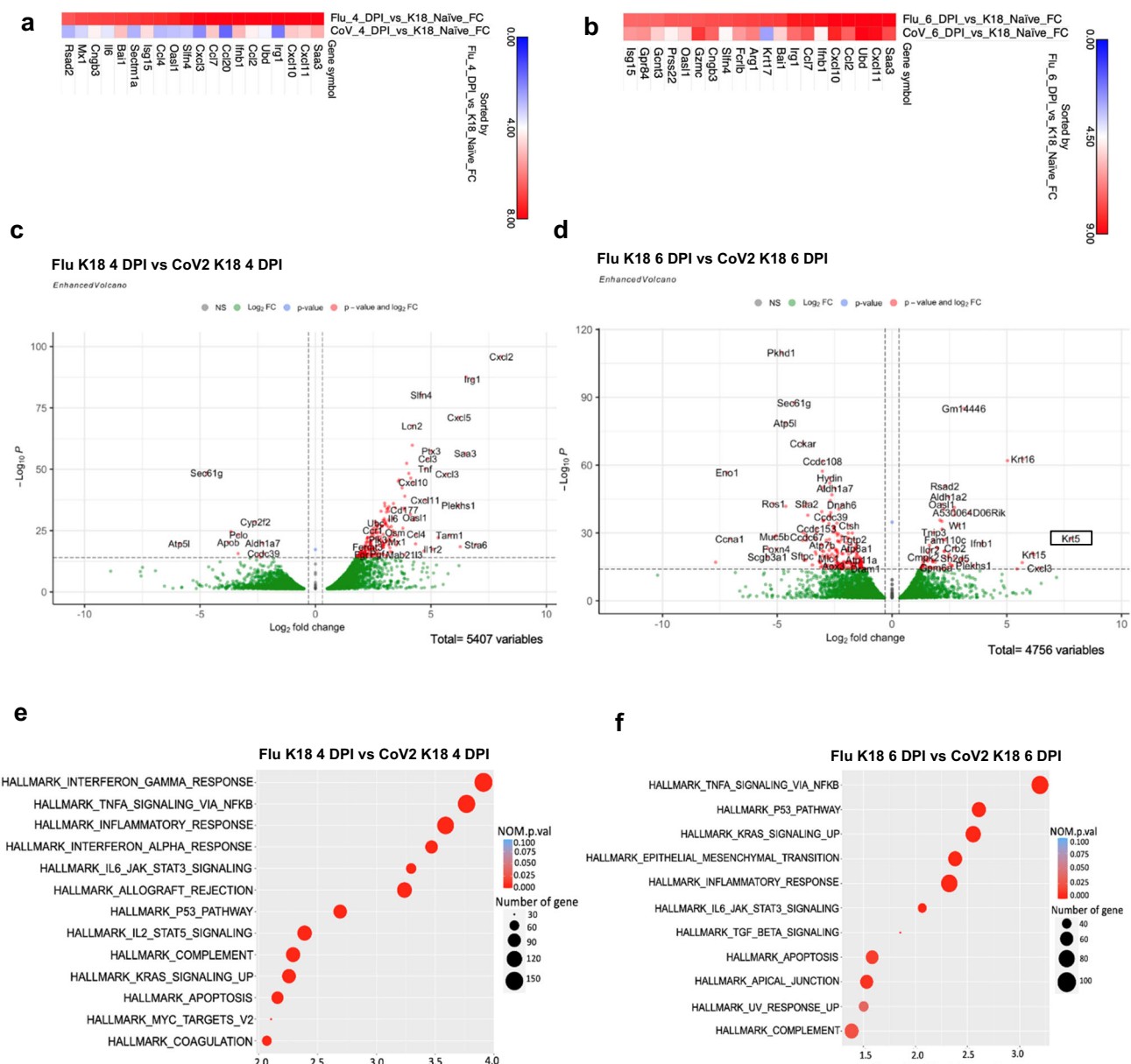

**Fig. 3 Comparison of gene expressions between Flu and SARS-CoV-2 infected lungs at day 4 and day 6 post infection.** Pulmonary RNA of $2 \times 10^5$ TCID50 dose-SARS-CoV-2 infected *K18-hACE2* mice and Influenza virus (50 PFU) infected B6 mice were collected and sequenced at 4 DPI ($n = 3$) and 6 DPI ($n = 3$). Heatmap shows the comparison of (**a**) Flu day 4 versus *K18* naïve and SARS COV2 day 4 versus *K18* naïve, gene sorted by Flu day 4 versus *K18* naïve **b** Flu day 6 versus *K18* naïve and SARS COV2 day 6 versus naïve *K18*, gene sorted by SARS COV2 day 6 versus naïve *K18*. The fold change threshold was 0–9. The top 20 genes were considered to generate all heatmaps. FC; fold change. **c, e** Enhanced volcano plot of DEGs (**a**) and pathways (**c**) between Flu (Flu *K18* 4 DPI) and SARS-CoV-2 (CoV2 *K18* 4 DPI) infected lungs at day 4 post infection. **d, f** Enhanced volcano plot of DEGs (**b**) and pathways (**d**) between Flu (Flu *K18* 6 DPI) and SARS-CoV-2 (CoV2 *K18* 6 DPI) infected lungs at day 6 post infection.

infection with CoV2 induces persistent fibrosis in the lungs of *K18* mice at 21 DPI.

**Krt5+ cell proliferation is absent in pulmonary consolidation area in CoV2 infected NHPs and human COVID-19 patients.** We validated our finding with necropsied lung samples from CoV2-infected NHPs (Supplementary Table 4), and three fatal human COVID patients (case summary shown in Supplementary Table 5 and Supplementary Fig. 10). We compared the COVID NHP lungs acutely infected with a different respiratory virus, respiratory syncytial virus (RSV), to provide a comparator for Flu infected mouse model. Figure 6a shows that the acute RSV-

infected NHPs at 7 DPI (left panel, $n = 1$) exhibited a strong Krt5 expression and cell proliferation in airways and in pulmonary parenchyma at site of consolidation with increased SMA expression. We previously reported that the CoV2-infected NHPs had typical COVID symptoms and pathological changes[31]. In acutely CoV2-infected NHPs between 14 and 30 DPI, Krt5 were only present in its normal niche within airways. We did not detect the proliferation of Krt5+ cells in regions of consolidation that displayed increased SMA expression within the pulmonary parenchyma (Fig. 6a, right panel, $n = 4$). Analysis of IF from three adult patients[23] showed a similar lack of Krt5+ cell proliferation and "pod" structure as well (patients 1 shown in Fig. 6b; and patients 2 and 3 shown in Supplementary Fig. 11). Using

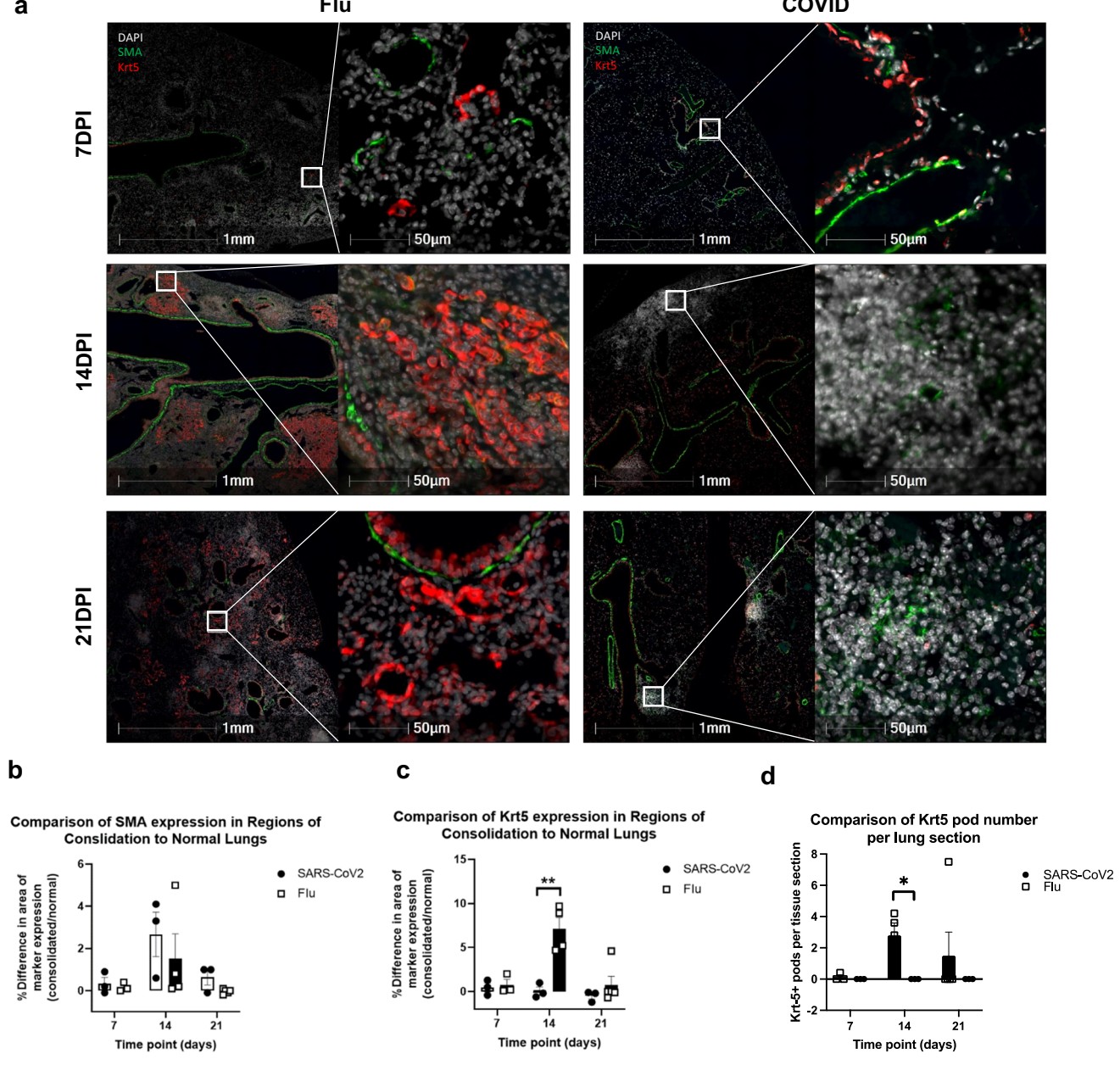

**Fig. 4 Krt5+ progenitor cell proliferation was induced in Flu infected *K18* mice but not in SARS-CoV-2-infected *K18* mice at 7, 14, and 21 DPI. a** SMA-Krt5 double staining shows proliferation of Krt5+ cells in consolidated regions of Flu-infected lungs at 7 ($n = 3$), 14 ($n = 4$) and 21 DPI ($n = 5$). In SARS-CoV-2 infected lungs, consolidated pulmonary regions do not exhibit proliferation of Krt5+ cells (red) at 7 ($n = 3$), 14 ($n = 3$), or 21 DPI ($n = 3$). Comparison of the expression of SMA (**b**) and Krt5 (**c**) in regions of consolidated and normal lungs from the same animal as detailed in Supplementary Fig. 9. **d** Number of pulmonary regions with proliferation of Krt5+ cells ("pods") per lung section. Data are shown as mean ± SEM. To compare values obtained from two groups, two-tailed unpaired Student's *t* test was performed. ** indicates $p < 0.01$. * indicates $p < 0.05$.

both NHPs and human samples, we demonstrated a lack of a Krt5+ progenitor cells response following acute CoV2 infection.

**scRNA-seq analysis of CoV2 and A/PR/8/34 (H1N1) infected *K18* mice at 4 DPI.** We further performed scRNA-seq analyses of flu-infected mice at an early phase of the infection (4 DPI). Pulmonary transcriptomic changes from Flu infection at the single-cell level were compared with those of CoV2 infection at 4 DPI, at the peak point of lung viral load when early pneumonia may appear[15]. The scRNA-seq analysis identified 10 distinct clusters of cells (Fig. 7a). We did not observe that *Krt5* expression was increased in any of the identified 10 cell populations in CoV2 or Flu-infected mice in lungs at 4 DPI until 6 DPI (Supplementary

Data 9). These results are consistent with the bulk RNA analysis of Flu and CoV2-infected mice at 4 and 6 DPI (Fig. 3a–d). The interferon-induced protein with tetratricopeptide repeats 1 (*Ifit1*), 2 (*Ifit2*), 3 (*Ifit3*), and the interferon-inducible chemokine *Cxcl10* were significantly upregulated in all eight cell populations of Flu-infected lungs compared to the CoV2 infected lungs (Fig. 7b–e). These findings confirm the results from bulk RNA analyses and further demonstrate that CoV2 infection induced down-regulation of the interferon response pathway on all pulmonary cells but not specific cell populations. We further investigated the impact of CoV2 and Flu infection on pulmonary epithelial cells specifically in the scRNA-seq data. Transcriptomic differences and pathway activation within the pulmonary epithelial cell cluster from Flu-

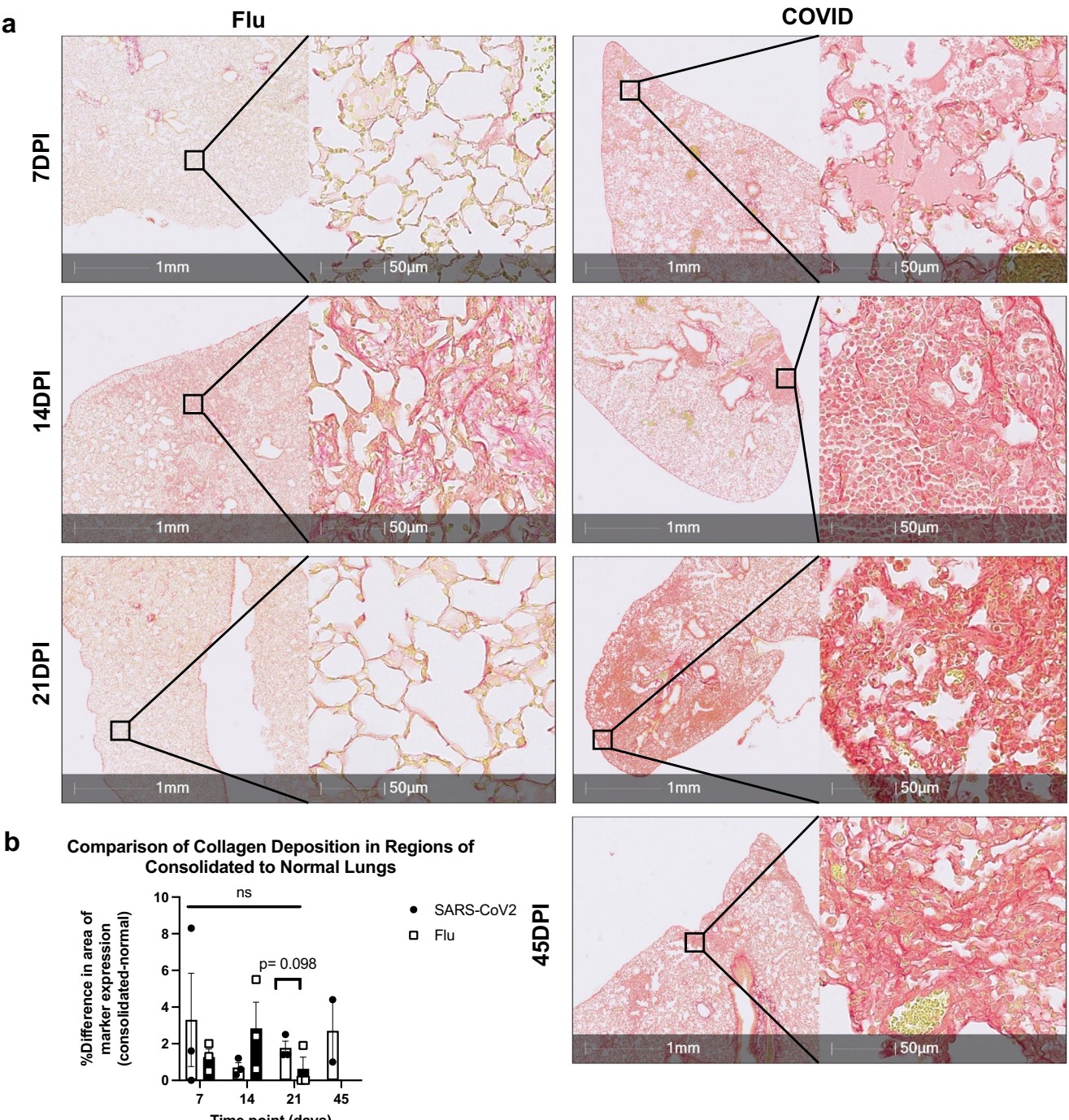

**Fig. 5 Collagen deposited in Flu mouse model and COVID mouse model. a** Representative image of PSR stainings showing collagen deposition in lungs of SARS-CoV-2-infected *K18* mice (IN, 1 × 10⁴ TCID₅₀) at 7 ($n = 3$), 14 ($n = 3$), 21 ($n = 3$) and 45 DPI ($n = 2$) and Flu infected mice at 7 ($n = 3$), 14 ($n = 4$) and 21 DPI ($n = 3$) (50 PFU). **b** Quantification of collagen deposition in regions of consolidated and normal SARS-CoV-2-infected and Flu-infected lungs as detailed in Supplementary Fig. 7. Data are shown as mean ± SEM. Two-way analysis of variance (ANOVA) was used to compare collagen deposition level changes from 7 DPI to 21–45 DPI. One-tailed unpaired Student's *t* test was performed to test the difference between two groups at one time point.

and CoV2-infected mice were compared at 4 DPI. There were 105 up- and 744 down-regulated genes in flu-infected pulmonary epithelial cells compared to in CoV2 infection. We observed significant up-regulation of several interferon-stimulated genes (ISGs), including *Cxcl10, Ifit2, Gbp2, Rsad2,* and *Isg15* in Flu-infected lung epithelial cells compared to CoV2-infected cells (Fig. 8a and Supplementary Data 8). Both TNF receptor superfamily member 12 A (*Tnfrsf12a*), which may be involved in positive regulation of the extrinsic apoptotic signaling pathway

and regulation of wound healing[32], and *Serpinb9*, which may be related to oxidation-reduction reactions[33], were upregulated in influenza epithelial cells only and not observed in CoV2 infected mice (Fig. 8a and Supplementary Data 8). Among the down-regulated genes in Flu infected cells compared to CoV2, *Sftpa1, Sftpb, Sftpc, and Sftpd*, four surfactant proteins produced by type II pneumocytes[34], and *Scgb1a1, Scgb3a1,* and *Scgb3a2*, three cytokine-like secretory proteins of small molecular weight secreted by lung epithelial cells that serve as biomarkers for lung injury[35]

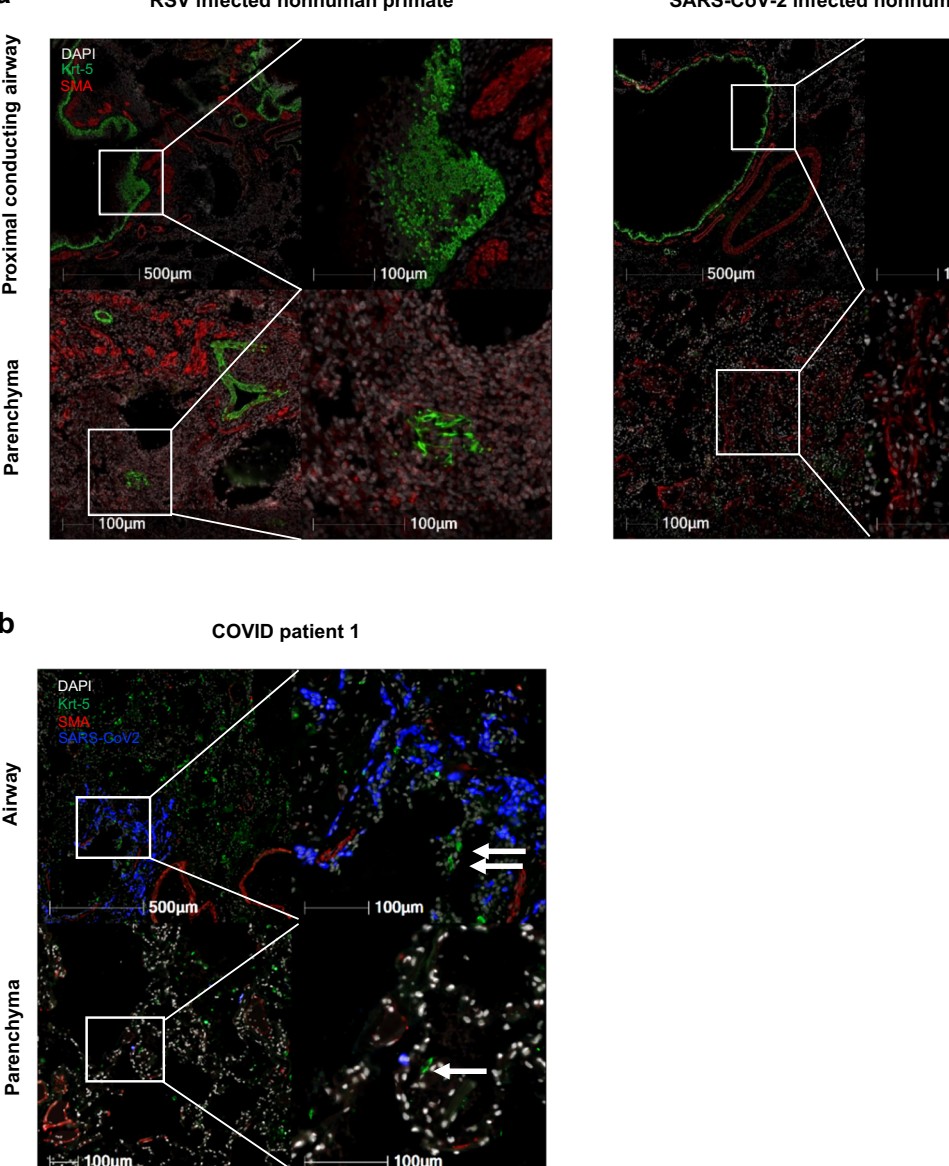

**Fig. 6 No Krt5$^+$ progenitor cells proliferate in CoV2-infected nonhuman primates or human patients. a** Krt5$^+$ progenitor cells (green) are normally identified in the basal layer of proximal conducting airways as seen in the SARS-CoV-2 infected NHP (top right). Following RSV infection in nonhuman primate (n = 1), there is proliferation of Krt5+ cells within airways and the adjacent pulmonary parenchyma which is not observed in SARS-CoV-2-infected nonhuman primates (n = 4). **b** Adult human COVID patient (patient 1) exhibits a similar reaction pattern to SARS-CoV-2 infected nonhuman primates with Krt5+ cells largely restricted to airways and rarely found within the pulmonary parenchyma.

were demonstrated in Flu-infected lungs. These down-regulated genes in Flu-infected lungs relative to CoV2 may reflect the extensive injury mediated by Flu infection. The interferon-gamma and alpha pathways, IL2-stat5 signaling pathway, apoptosis pathway, TNF α signaling via NFκB pathway, hypoxia pathway, UV response pathway, and P53 pathway were increased in the epithelial cells of Flu infected lungs compared to CoV2-infected lungs (Fig. 8b). These genes and pathways identified may contribute to the distinct lung injury and induced Krt5 progenitor cell proliferation between Flu and COVID infection-mediated lung.

## Discussion
A major limitation in understanding long-term COVID disease has been a lack of small animal models. *K18* mice are on the B6 background and have been widely used to mimic acute severe

COVID-19 for pathogenesis studies[36,37] and preclinical measurements for assessing anti-COVID-19 therapeutics and vaccines. Our study demonstrates that infection of *K18* mice with a sub-lethal concentration of CoV2 induces persistent pathology out to 45 DPI, consistent with long COVID. The mouse adapted CoV2 strain—MA10—has recently been shown to replicate in upper and lower airways of both young adult and aged mice and can model both acute and chronic COVID[38]. However, MA10 only causes evident COVID in BALB/c mice[38], preventing the usage of many genetic tools available on C57BL/6 background mice. The golden hamster has also been utilized as a small animal model for long COVID, showing unique permanent injuries in lung and kidney[39]. However, limited genetic information and tools (specific biological reagents) preclude this model for extensive mechanistic study as well. Our findings demonstrate

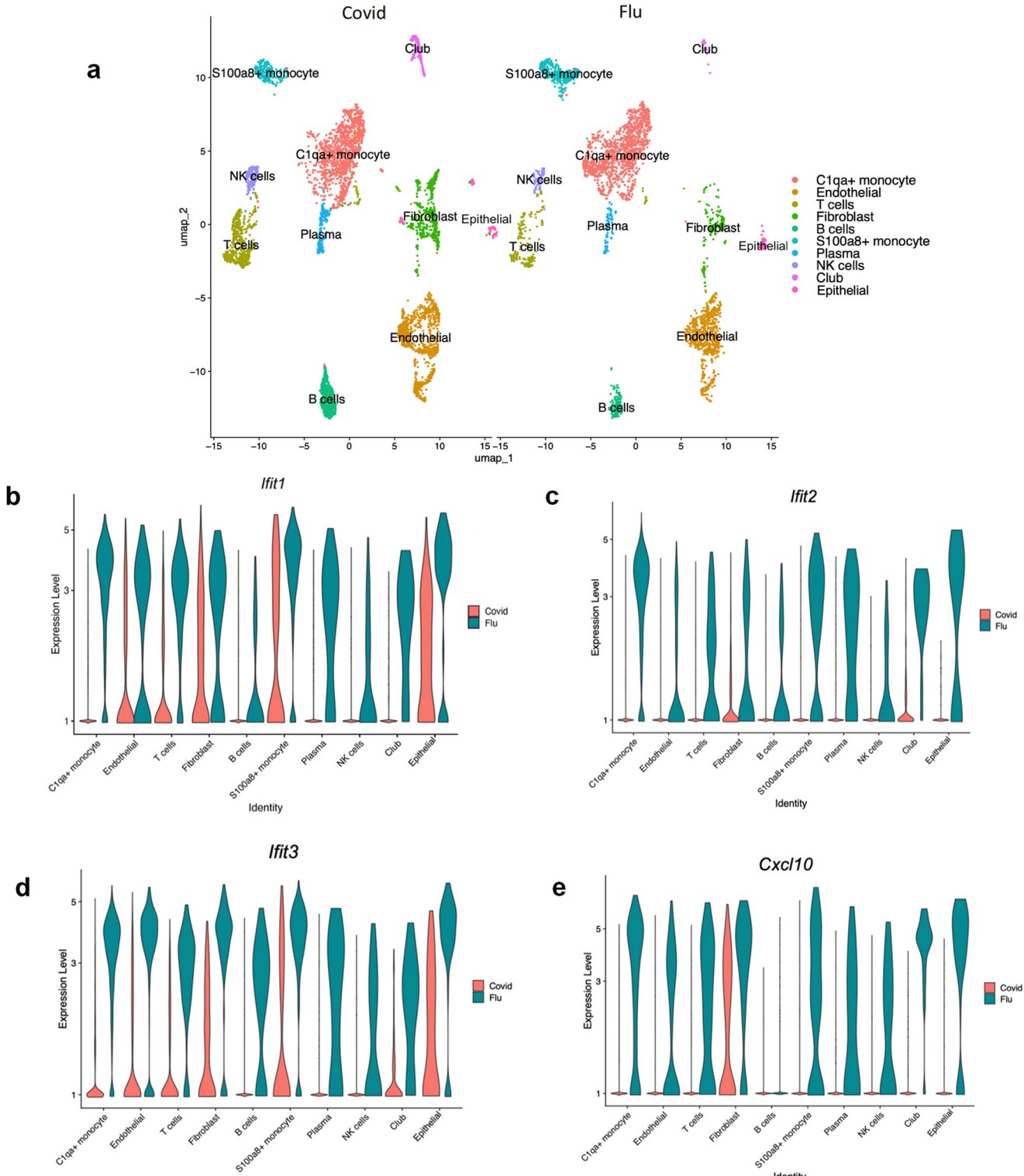

**Fig. 7 scRNA-seq analysis of Flu-infected and SARS-CoV-2-infected lungs. a** Major clusters and respective cell types for Flu and SARS-CoV-2 tissues at 4 DPI by scRNA-seq data. Uniform manifold approximation and projection (UMAP) for dimension reduction plot with major cell types of scRNA-seq. Single-cell suspensions from whole infected lungs at 4 DPI for both SARS-CoV-2 and Flu were processed and sequenced. The 8146 cells identified after being combined and processed, we identified ten major clusters including C1qa+ monocyte ($n = 2,482$), Endothelial ($n = 1637$), T cells ($n = 892$), Fibroblast ($n = 804$), S100a8+ monocyte ($n = 633$), Plasma ($n = 280$), NK cells ($n = 258$), Club ($n = 220$), and Epithelial ($n = 207$). **b**–**e** Expression of *Ifit1, Ifit2, Ifit3*, and *Cxcl10* in the Fu-infected and SARS-CoV-2-infected lungs.

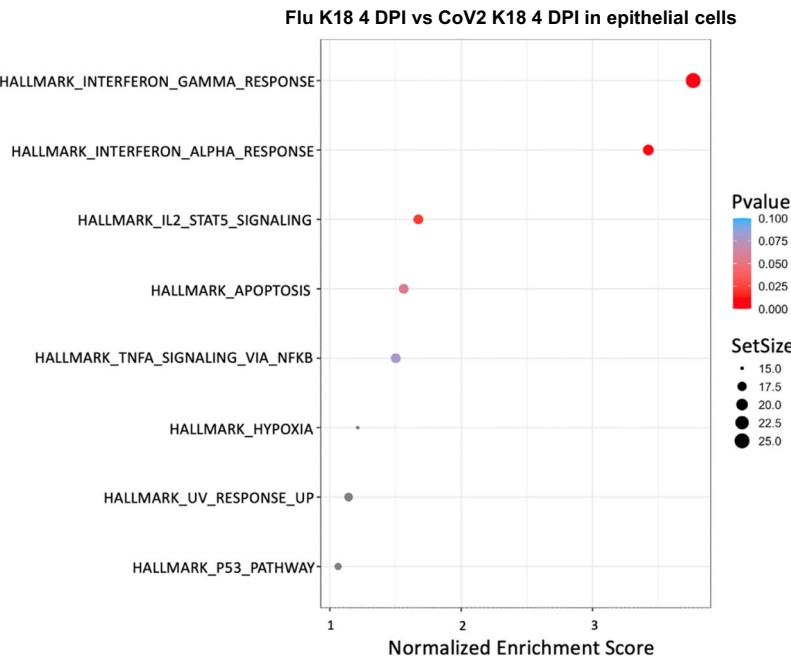

**Fig. 8 scRNA-seq analysis of Flu-infected and SARS-CoV-2-infected pulmonary epithelial cells. a** Enhanced volcano plot for 4 DPI epithelial cells cluster DE genes. **b** Significant pathway for 4 DPI Flu-infected versus SARS-CoV-2-infected lungs (epithelial cells cluster).

*K18* mice on B6 background as a valid model to investigate pathogenesis of CoV2 and mechanisms of long COVID.

The persistent, patchy lesions found in murine lung at 21 and 45 DPI appear to be independent of live virus, as we were unable to detect viral subgenomic RNA at these late timepoints. However, we did identify CoV2 spike protein in pulmonary macrophages. This data indicate that the CoV2 may be engulfed and cleared by pulmonary macrophages at 21 DPI. Previously, we found that infiltrating macrophages are the predominant cell population in response to CoV2 infection in *K18* mice during the acute phase via scRNA seq analysis[15]. Dense infiltration of aberrantly activated

monocyte-derived macrophages and alveolar macrophages have been described in inflamed lungs from severe COVID-19 patients[40]. Together, our data further highlight the important role of macrophages in the pathogenesis of acute and chronic COVID.

We also documented that influenza induced significantly stronger inflammatory responses within 4 DPI compared to CoV2. One likely reason for this heightened inflammation during acute Flu infection is the stronger interferon response, as evidenced by the induction of interferon stimulated genes (ISGs) including *Ifit1, Ifit2, Ifit3,* and *Cxcl10*. These differences in interferon pathway induction may trace back to TLR3 and

TLR7 sensitivity or other IFN stimulating and regulatory factor differences. It has also been shown that CoV2 spike protein directly suppresses interferon regulatory factor 3 (IRF3), a key transcriptional regulator of Type I interferons[41] .

The interferon response is the most extensive and immediate innate immune response during viral infections. While IFNs are induced during many viral infections[42], their role during COVID has been somewhat controversial with conflicting findings. Initial reports in mice infected with CoV2 suggested IFN is a contributing factor to excessive inflammation[43]; however, administration of Type I IFNs reduced viral titers and improved recovery in a CoV2-infected animal model. It is important to note that IFN is only effective when administered at early phase after infection[44,45], and sustained Type I IFN levels have been associated with increased pulmonary injury and reduced epithelial cell proliferation[42]. Importantly, sustained IFN production has been verified in mild-to-moderate long COVID patients at 8 months after disease[46]. In our study, we also detected sustained IFN pathway activation at 21 DPI in the lungs of COVID mice. However, how the sustained IFN pathway activation contributes to persistent fibrosis, lack of induction of Krt5+ "pod" formation and cell proliferation in CoV2-infected lung remains unclear and requires further investigation.

Our bulk RNAseq findings demonstrated significant transcriptional differences between Flu and CoV2-infected mice. While a number of these genes were associated with inflammation, key differences in several keratin genes were also identified. Specifically, there was a significant induction of *Krt5* following Flu infection, which is of particular interest as it is a marker for lung multipotent basal cells. Basal epithelial cells, locate on the basal side of pseudo-stratified epithelia[47,48], have been characterized to expand and migrate from the conducting airways to injury sites during Flu infection[12,49,50]. While initially thought to replace dying epithelial cells, there is a growing consensus that basal cells act to prevent loss of structural integrity during injury[51]. Histologically, we demonstrated nascent progenitor cells formed "pod" structure in pulmonary parenchyma in Flu but not CoV2-infected mice stained by Krt5, trp63, and NGFR. The Krt5+ "pods" mainly contain AT1 cells, which confirms the notion that nascent Krt5+ cells may contribute to repair following Flu-induced pulmonary injury. Unlike Flu, Krt5+ cells were not involved in the response following CoV2-induced pulmonary injury in mice, NHPs, and human lungs. Until now, there have been a very limited number of publications investigating basal cells and CoV2 infection. In a recent clinical study, Jyothula et al. tested the lungs collected from the COVID-19 patients who required transplantation due to the fulminant lung fibrosis[52]. The study identified a unique fibrotic gene signature in these patients dominated by a hyper-expression of pro-fibrotic genes and found an increase of Krt-5 expressing cells adjacent to fibroblastic areas and in apparent epithelial bronchiolization areas[52]. Of note, these patients suffered from non-resolvable lung fibrosis for at least two months. Most of these patients received steroids (dexamethasone) and remdesivir as well as being placed on invasive mechanical ventilation or ECMO prior to transplant. The multiple extensive clinical interventions, treatments, and prolonged COVID-19 disease course may have induced the Krt5+ cell proliferation observed in their study. In contrast, our study investigated the natural disease course COVID in the absence of clinical intervention in which we failed to detect Krt5+ proliferating cells in fibrotic areas in *K18* mice, NHP, and human COVID samples within approximately one month of infection, a period that spans the transition from the acute to the chronic phase. It is conceivable that the inability for CoV2 to induce Krt5+ cells during the acute phase of disease may be a key factor for the protracted COVID injury outcomes.

Interestingly, we also found that COV2-infected *K18* mice had increased AT2 cell numbers compared to Flu-infected mice at 21 DPI (Supplementary Fig. 12). Another function of AT2 cells is to express pulmonary surfactants. Consistently, our sc RNA analysis documents that the up-regulation of surfactant genes such as Sftpa Sftpb, Sftpc, and Sftpd in the epithelial cells of COV2 infected K18 as compared to Flu-infected K18 mice at 4DPI (Fig. 8a). However, how AT2 cells, another group of stem cells in the lung[53], contribute to repair of chronic COVID-mediated lung injury and normalization of lung function remains unclear. Nonetheless, our scRNA analysis in Flu and CoV2 infected lung identified key molecular and pathway activation differences in epithelial cells. The causative effect of these genes and pathways on Krt5+ "pod" induction and lung repair in viral infection-mediated chronic pulmonary injury remains elusive.

Taken together, by comparing COVID-19 infection vs Influenza infection in *K18* mice, we documented that 1) CoV2 infection but not Flu infection fails to induce Krt5+ "pod" formation and cell proliferation and 2) CoV2 infection mediates more profound chronic effects on the lungs including fibrotic abnormalities than flu infection. Our studies provide a new avenue for further identifying cellular and molecular differences in the repair process between COVID-19 and influenza. It is important to note that there are many differences between Flu and COV2 infections. Influenza is a segmented genome and induces a stronger IFN response, can replicate faster as compared to SARS-COV-2, and both viruses use different receptors to cause the diseases[54–57]. How those differences such as growth rates and the utilization of the different receptors influence the resolution of the inflammation and fibrotic abnormalities mediated by Influenza and SARS-COV-2 is unknown and warrants further investigation experimentally. The dissection of the questions requires the use of multiple approaches including in vitro cell culture system[54], human organoid method[58,59], our model reported here, other mouse-adapted SARS-CoV2 strains such as MA10 or MA30, and human Flu and COVID patient samples[56,60,61].

## Methods

**Mice**. *K18*+/−(034860) and *C57BL/6J* (*B6*) mice were purchased from the Jackson Laboratory and housed in an animal facility at Tulane University (detailed mice information are shown in Supplementary Table 1).

**Statistics and reproducibility**. The results presented depict the output of independent mice ($n = 1–5$). Infected mice were analyzed at multiple time intervals post-infection to minimize bias and increase reproducibility. Data are shown as mean ± SEM. To compare values obtained from multiple groups over time, two-way analysis of variance (ANOVA) was used, followed by Bonferroni post hoc test. To compare values obtained from two groups, the unpaired Student's *t*-test was performed. Statistical significance was taken at the $P < 0.05$ level.

**Study approval**. All animal experiments were reviewed and approved by the Institutional Animal Care and Use Committee at Tulane University and conducted under the Animal Use Protocols: 1331 and 1383. We have complied with all relevant ethical regulations for animal studying.

**SARS-CoV-2 and influenza infection**. For SARS-CoV-2, the USA-WA1/2020 isolate was used (NR-52281 strain deposited by the Centers for Disease Control and Prevention and obtained through BEI Resources, NIAID, NIH). We passaged the virus in VeroE6-TMPRSS2 cells in DMEM media with 2% FBS and

sequence verified to ensure homology with the original patient sequence. Mice were infected intranasally with SARS-CoV-2 ($2 \times 10^5$ TCID50 for acute lethal dose infection or $1 \times 10^4$ TCID50 for sub-lethal dose infection) in an ABSL3 facility.

For influenza (Flu) infection, mice were infected with H1N1 A/PR/8/34 (PR8)[6,27,62]. Briefly, a sublethal concentration (50PFU) was diluted in 50ul of sterile saline and mice were infected by oropharyngeal aspiration. Control, uninfected mice received sterile saline.

**Tissue processing**. ABSL3 facility staff monitered the daily body weights. The mice were euthanized if the mice lose 20% body weight. Half lung (whole Right lung) was fixed inside Z-FIX buffer at RT. Half of the rest half lung (Left) were submerged in 1 mL Trizol and stored at $-80\,°C$ for RNA extraction. The remaining half of the rest half lung (Left) were frozen on dry ice with no media store at $-80\,°C$ as backup. Brain was collected 1/4 in flash freeze on dry ice with no media store at $-80\,°C$,1/4 in 1 mL Trizol for RNA extraction, 1/2 in Z-fix for histological analysis.

**Histological analysis and quantification of histopathologic lesions**. Sections of the lung were processed routinely, stained with hematoxylin and eosin (H&E), and digitally scanned with a Axio Scan.Z1(Zeiss, Thornwood, NY). Images were captured using the Figure maker tool in HALO (Indica Labs, Albuquerque, NM).

**RNA isolation**. Tissues were collected in 1 mL Trizol reagent (15596026; Invitrogen) and extracted with RNeasy Mini Kit (Cat. No.74104; QIAGEN, Hilden, Germany) following the manufacturer's protocol. The concentration of RNA was determined by NanoDrop 2000.

**Subgenomic N viral copy number detection**. 100 ng total RNAs were mixed in Taqpath 1-Step Multiplex Master Mix (Cat. No. A15299; Thermo Fisher, Waltham, MA) and FAM-labeled primers (sgm-N-FOR: 5′-CGATCTCTTGTAGATCTGTTCTC-3′, sgm-N-Probe:5′-FAM-TAACCAGAATGGAGAACGCAGTGG G-TAMRA-3′,sgm-N-reverse:5′-GGTGAACCAAGACGCAGTA T-3′), following the manufacturer's instructions. Subgenomic N viral copy number was calculated by standard Cq values. The assay was performed under ABI QuantStudio 6 system (Thermo Fisher, Waltham, MA).

**Bulk RNA sequencing**. Isolated lung tissue RNA was quantified using Qubit 3.0 Fluorometer (ThermoFisher Scientific, Waltham, MA). Agilent 4150TapeStation (Santa Clara, CA) was used to obtain RNA integrity number and fragment sizes (DV200 metrics). Illumina NEBNext Ultra II directional RNA library prep kit (San Diego, CA) was used for library preparation. For cluster generation, the cDNA libraries were pooled at a final concentration of 1.8 pM. Illumina NextSeq 2000 P2 100 (San Diego, CA) was used with a minimum of 750 pM of RNA for sequencing. Gene expression and nucleotide variation were evaluated after processing and mapping the raw reads as previously described[54]. We normalized the raw read counts across all samples. Differential expression analysis was performed using Cuffdiff, EdgeR, and DESeq, (Slug Genomics, University of California, Santa Cruz). To generate volcano plots in R, the Cuffdiff output was considered. All curated datasets were deposited in the Sequence Read Archive BioProject number: GSE248778.

**Volcano plot**. R software v4.1.0 package EnhancedVolcano was used to generate volcano plots. The x-axis coordinate was log2 fold change and the y-axis coordinate was negative log10 p value.

**Over-enrichment analysis**. Enrichment analysis was performed with the R package version 4.1.0 software package ggplot2. Enrichment scores were calculated against the Hallmark (H) gene sets of the Molecular Signature database (MsigDB) database. Gene sets significantly enriched in the datasets ($p < 0.05$) were curated.

**Single-cell RNA sequencing and 10× assay**. Lung will be minced with forceps and small scissors and digested in 2 mL serum-free medium with 2 mg/mL collagenase (MilliporeSigma) and 80 U/mL DNase I (MilliporeSigma) for 60 min at 37 °C[15]. 5000 live cells per sample were targeted by using 10× Single Cell RNAseq technology provided by 10× Genomics (10X Genomics)[15]. Full-length barcoded cDNAs were then generated and amplified by PCR to obtain sufficient mass for library construction[15]. Pooled libraries at a final concentration of 1.8 pM were sequenced with paired-end single index configuration by Illumina NextSeq 550. Cell Ranger version 2.1.1 (10X Genomics) was used to process raw sequencing data and Loupe Cell Browser (10X Genomics) to obtain differentially expressed genes between specified cell clusters[15]. In addition, Seurat suite version 2.2.1 was used for further quality control and downstream analysis[15]. Filtering was performed to remove multiplets and broken cells, and uninteresting sources of variation were regressed out. Variable genes were determined by iterative selection based on the dispersion versus average expression of the gene. For clustering, principal component (PC) analysis was performed for dimension reduction. The top 10 PCs will be selected by using a permutation-based test implemented in Seurat and passed to t-SNE for clustering visualization[15].

**Immunostaining**. Zinc formalin-fixed (Z-fixed), paraffin-embedded lung was sectioned, deparaffinized, and subjected to heat-induced epitope retrieval using both high-pH(Vector Labs H-3301, Olean, NY), and low-pH solutions (Vector Labs H-3300, Olean, NY). Sections were blocked with 10% goat serum or 1% donkey serum for 40 min at room temperature, incubated with the primary antibodies overnight at 4 °C and secondary antibodies for 60 min at room temperature. Slides were digitally scanned by Zeiss Axio scan. Z1. Detailed antibody information is shown in Supplemental Table 6. PSR staining and Trichrome staining were performed routinely by the histology core in TNPRC.

**Digital image analysis for the quantification of fluorescent marker expression and trichrome**. Smooth muscle actin (SMA) and Cytokeratin 5 (Krt5) were quantified using the Area Quantification Fl module v.2.1.10 in HALO v3.4 (Indica Labs, Albuquerque, NM). Sections of lung were stained with a panel of markers including DAPI, CK5, and SMA. Slides were imaged in 4 channels, including one empty channel (Far red) for the detection and subsequent exclusion of autofluorescent signal. A pathologist annotated regions of interest (ROI) for pulmonary lesions and adjacent regions of normal/unaffected lung. If multiple regions of pneumonia were present, those with the highest CK5 expression were chosen for quantitation. Thresholds for marker detection were set and checked for accuracy post analysis by the same pathologist. Each marker was quantified based on the percentage of area it was detected within each region (lesion or normal). Trichrome stain was quantified on the same ROI, using the same method but with Area Quantification v.2.2.4 module from HALO.

Quantification of colocalization of CD206 and SARS-CoV-2 was performed using the HighPlex FL v4.1.3 module in HALO. Slides were stained with DAPI, CD206, and SARS-CoV-2 and imaged in four channels, including one empty channel for autofluorescence detection. Region of interest was drawn around the entire lung section. Thresholds for marker detection were set and checked for accuracy post analysis by the same pathologist.

Trichrome and Picosirius red staining were quantified using deep learning, pattern recognition algorithms (HALO AI v3.4). Algorithms were trained by a pathologist to recognize collagen from both Trichrome and Picrosirius red stained slides. Regions of interest were drawn around regions of pulmonary consolidation and adjacent, similar, normal lung regions excluding anatomic regions that normally contain abundant collagen (large vessels and airways). Data was graphed as the difference in the percentage area of collagen in consolidated regions minus normal regions.

**Reporting summary**. Further information on research design is available in the Nature Portfolio Reporting Summary linked to this article.

## Data availability
Bulk RNA and Single-cell RNA sequencing data generated by this study have been deposited to the GEO database and will be available publicly at the date of publication (accession number: pending). The source data are included in the supplemental material or available upon request to the first authors or corresponding authors.

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

## Acknowledgements

This work was supported by NIH 5 P51OD011104-58 (C.W., Z.C., C.R.E., J.M.C., V.T.D., J.R., N.J.M., R.B., X.Q.), AHA962950 (X.Q.), R01DK129881 (X.Q.), R01HL165265 (X.Q.), R35 HL139930 (J.K.K.), the Louisiana Board of Regents Endowed Chairs for Eminent Scholars Program, Emergent Ventures-Fast Grant, and R35 HL139930 (J.K.K.), and Tulane start-up funds (X.Q. and J.R.). The following reagents were deposited by the Centers for Disease Control and Prevention and obtained through BEI Resources, NIAID, NIH: SARS-Related Coronavirus 2, Isolate USA-WA1/2020, NR-52281 and polyclonal anti-SARS coronavirus (antiserum, Guinea Pig), NR-10361. We thank Haiyan D Miller and Kejing Song for technical assistance related to bulk RNA analyses and NGS sequencing and Angela Birnbaum, Tammy P Bavaret, and Solange Paredes with technical assistance related to BSL3 experiments. We also thank Cecily C Midkiff, confocal microscopy and molecular pathology core at Tulane National Primate Research Center for technical assistance related to immunostaining, and Kelly Goff and Sarah Scheuermann, VCIPS core at the TNPRC, for expanding, characterizing, and providing the SARS-CoV-2 used in these studies.

## Author contributions

J.K.K., D.A.P., and X.Q. developed concept. C.W., M.S.K., Z.Z., M.J.A., Z.C., C.R.E., J.M.C., G.D., R.V.B., N.J.M., J.K.K., and X.Q. contributed to performing the experiments and analyzed the results. R.V.B., V.T.D., and J.R. provided NHP data. D.T., K.B., and X.Y. provided human data. C.W., M.S.K., Z.Z., M.J.A, Z.C., C.R.E., J.M.C., R.V.B., V.T.D., J.R., N.J.M., R.V.B., J.K.K., D.A.P., and X.Q. wrote the manuscript and all authors participated in the review and critique of the manuscript. R.V.B., J.K.K., D.A.P., and X.Q. interpreted the results and supervised the experiments.

## Competing interests

The authors declare no competing interests.
