## [Peer Review File · Communications Biology]

Reviewers' comments:

Reviewer #1 (Remarks to the Author):

The article by Chenxiao Wang et al explored a possible mechanism of post-COVID-19 respiratory sequelae focusing on the abnormal repair. This study analyzed data from the sub-lethal SARS-COV-2 infections in K18-ACE2 transgenic mice and the mouse-influenza model. Compared to influenza infection, SARS-COV-2 infected mice showed loss of Krt5+ cells with persistent interstitial collagenous depositions until 21 days post-infection (dpi), while influenza-challenged mice show induction of Krt5+ expressing cells. Further, ScRNA sequencing exhibited low interferon expression in SARS-COV-2 infection. Histopathologic findings demonstrate persistent pathology until 21 dpi in SAR-COV-2, while influenza-challenged mice recover to normal re-epithelialization by 14 dpi.

Minor concerns:

Although influenza and COVID-19 share significant similarities, in ARDS pathophysiology, distinct fibrotic abnormalities have been highlighted more frequently in SARS-COV-2 infections, compared to influenza. The current study thus may open a new avenue in identifying differences in the repair process between COVID-19 and influenza. The authors should consider using more stringent models and pathologies to understand the tissue repair in these viral infections. Influenza and SARS-COV-2 have different growth rates. Influenza replication is much faster than SARS-COV-2, the authors should consider evaluating whether the growth rate influence abnormal fibrotic tissue remodeling?

Reviewer #2 (Remarks to the Author):

Please brushup the following sentences.

The authors have appropriately addressed most of the comments. Thank you for providing additional data. I believe this paper is nearing the level of acceptance by Commun Biol. I have some minor comments concerning the immunostaining images.

1. The images in Figure S2 are of low magnification. Preferably, higher-magnification images should be used.
2. It is difficult to confirm the DAPI signal in Figure S4. Why does P63 not appear to co-localize with the nucleus?

Point by Point response to the Reviewers' comments:

Reviewers' comments:

Reviewer #1 (Remarks to the Author):

Minor concerns: Although influenza and COVID-19 share significant similarities, in ARDS pathophysiology, distinct fibrotic abnormalities have been highlighted more frequently in SARS-COV-2 infections, compared to influenza. The current study thus may open a new avenue in identifying differences in the repair process between COVID-19 and influenza. The authors should consider using more stringent models and pathologies to understand the tissue repair in these viral infections. Influenza and SARS-COV-2 have different growth rates. Influenza replication is much faster than SARS-COV-2, the authors should consider evaluating whether the growth rate influence abnormal fibrotic tissue remodeling?

Response: Thank you for the insightful comments. To address this concern, we have added the following discussion in the manuscript: "Taken together, by comparing COVID-19 infection vs Influenza infection in K18 mice, we documented that 1) CoV2 infection but not Flu infection fails to induce of Krt5+ "pod" formation and cell proliferation and 2) CoV2 infection mediates more profound chronic effects on the lungs including fibrotic abnormalities than flu infection. Our studies provide a new avenue for further identifying cellular and molecular differences in the repair process between COVID-19 and influenza. It is important to note that there are many differences between Flu and COV2 infections. Influenza is a segmented genome and induces a stronger IFN response, can replicate faster as compared to SARS-COV-2, and both viruses use different receptors to cause the diseases (54-57). How those differences such as growth rates and the utilization of the different receptors influence the resolution of the inflammation and fibrotic abnormalities mediated by Influenza and SARS-COV-2 is unknown and warrants further investigation experimentally. The dissection of the questions requires the use of multiple approaches including in vitro cell culture system(54), human organoid method(58, 59), our model reported here, other mouse adapted SARS-CoV2 strains such as MA10 or MA30, and human Flu and COVID patient samples(56, 60, 61)."

Reviewer #2 (Remarks to the Author):

The authors have appropriately addressed most of the comments. Thank you for providing additional data. I believe this paper is nearing the level of acceptance by Commun Biol. I have some minor comments concerning the immunostaining images.

Minor 1 comment. The images in Figure S2 are of low magnification. Preferably, higher-magnification images should be used.

Response: Thank you so much for your great comments. Based on your suggestion, we have added higher magnification images accordingly.

2. It is difficult to confirm the DAPI signal in Figure S4. Why does P63 not appear to colocalize with the nucleus?

Response: Thank you so much. Based on your suggestion, we have added additional Supplemental Figure 4B to show that the Trp63 signals colocalize with the DAPI signals.